# Melatonin Delays Arthritis Inflammation and Reduces Cartilage Matrix Degradation through the SIRT1-Mediated NF-κB/Nrf2/TGF-β/BMPs Pathway

**DOI:** 10.3390/ijms25116202

**Published:** 2024-06-04

**Authors:** Mingchao Zhao, Di Qiu, Xue Miao, Wenyue Yang, Siyao Li, Xin Cheng, Jilang Tang, Hong Chen, Hongri Ruan, Ying Liu, Chengwei Wei, Jianhua Xiao

**Affiliations:** Heilongjiang Key Laboratory Animals and Comparative Medicine, College of Veterinary Medicine, Northeast Agricultural University, Harbin 150030, China; 17645474939@163.com (M.Z.); qd13836005047@163.com (D.Q.); 13204076720@163.com (X.M.); 13845494535@163.com (W.Y.); dylisiyao@163.com (S.L.); chengx0606@163.com (X.C.); kuyurentang@163.com (J.T.); ch980815molihua@163.com (H.C.); ruanhongri@163.com (H.R.); 15665630021@163.com (Y.L.); neauweiwei@126.com (C.W.)

**Keywords:** chondrocyte, Melatonin, SIRT1, NF-κB, TGF-β, BMPs

## Abstract

Cartilage, a flexible and smooth connective tissue that envelops the surfaces of synovial joints, relies on chondrocytes for extracellular matrix (ECM) production and the maintenance of its structural and functional integrity. Melatonin (MT), renowned for its anti-inflammatory and antioxidant properties, holds the potential to modulate cartilage regeneration and degradation. Therefore, the present study was devoted to elucidating the mechanism of MT on chondrocytes. The in vivo experiment consisted of three groups: Sham (only the skin tissue was incised), Model (using the anterior cruciate ligament transection (ACLT) method), and MT (30 mg/kg), with sample extraction following 12 weeks of administration. Pathological alterations in articular cartilage, synovium, and subchondral bone were evaluated using Safranin O-fast green staining. Immunohistochemistry (ICH) analysis was employed to assess the expression of matrix degradation-related markers. The levels of serum cytokines were quantified via Enzyme-linked immunosorbent assay (ELISA) assays. In in vitro experiments, primary chondrocytes were divided into Control, Model, MT, negative control, and inhibitor groups. Western blotting (WB) and Quantitative RT-PCR (q-PCR) were used to detect Silent information regulator transcript-1 (SIRT1)/Nuclear factor kappa-B (NF-κB)/Nuclear factor erythroid-2-related factor 2 (Nrf2)/Transforming growth factor-beta (TGF-β)/Bone morphogenetic proteins (BMPs)-related indicators. Immunofluorescence (IF) analysis was employed to examine the status of type II collagen (COL2A1), SIRT1, phosphorylated NF-κB p65 (p-p65), and phosphorylated mothers against decapentaplegic homolog 2 (p-Smad2). In vivo results revealed that the MT group exhibited a relatively smooth cartilage surface, modest chondrocyte loss, mild synovial hyperplasia, and increased subchondral bone thickness. ICH results showed that MT downregulated the expression of components related to matrix degradation. ELISA results showed that MT reduced serum inflammatory cytokine levels. In vitro experiments confirmed that MT upregulated the expression of SIRT1/Nrf2/TGF-β/BMPs while inhibiting the NF-κB pathway and matrix degradation-related components. The introduction of the SIRT1 inhibitor Selisistat (EX527) reversed the effects of MT. Together, these findings suggest that MT has the potential to ameliorate inflammation, inhibit the release of matrix-degrading enzymes, and improve the cartilage condition. This study provides a new theoretical basis for understanding the role of MT in decelerating cartilage degradation and promoting chondrocyte repair in in vivo and in vitro cultured chondrocytes.

## 1. Introduction

Osteoarthritis (OA) is influenced by a variety of factors, including genetics, mechanical injury, aging, and metabolic syndrome. The degeneration of articular cartilage is an important manifestation of OA pathology [1]. As chondrocytes undergo aging or endure trauma, the degradation of cartilage ensues through the influence of cartilage-degrading enzymes (matrix metalloproteinases (MMPs), A disintegrin, and metalloproteases with thrombospondin motifs (ADAMTs)) and inflammatory cytokines (Interleukin-1β (IL-1β) and tumor necrosis factor (TNF)), leading to the degradation of aggregating proteoglycans and type-II collagen (COL2A1). This results in disrupted cartilage and synovial metabolic function [2,3]. In the context of synovial joints, cartilage damage precipitates synovial inflammation, impacting chondrocytes through the production of inflammatory cytokines and ECM-degrading enzymes by the inflamed synovium, thereby disrupting cartilage homeostasis and instigating a vicious cycle [4,5]. In a normal joint, joint cartilage, calcified cartilage, the subchondral bone plate (SBP), and the subchondral trabecular bone (STB) collaboratively form the “shock absorber”, responsible for load support and absorption [6]. Subchondral bone exhibits high sensitivity to pressure and damage, enabling the swift replacement of damaged bone with new bone through bone remodeling, thus establishing a stable physiological state [7].

MT possesses anti-inflammatory properties, regulates bone metabolism, inhibits osteoclasts, and exerts antioxidant effects [8,9,10]. In human chondrocytes, MT hinders the expression of induced nitric oxide synthase (iNOS) and cyclooxygenase 2 (COX-2) proteins and mRNAs, which are stimulated by H_2_O_2_, consequently diminishing the levels of downstream products such as nitric oxide (NO) and prostaglandin E2 (PGE2) [11,12].

Silent information regulator transcript-1 (SIRT1) is a histone deacetylase involved in fundamental physiological processes such as aging, energy regulation, metabolism, and inflammation [13]. SIRT1 collaborates with various pathways, including NF-κB/AMPK/Nrf2, to attenuate the expression of chondrocyte matrix-degrading enzymes induced by IL-1β and decelerate cellular degradation [14,15]. Furthermore, SIRT1 can regulate the differentiation of bone marrow mesenchymal stem cells into chondrocytes through Sry-related HMG box-9 (SOX9), promoting cartilage repair [16,17]. Previous experiments have demonstrated that MT can exert anti-inflammatory and protective effects in chondrocytes through the SIRT1 pathway [11], emphasizing the indispensability of SIRT1 in maintaining articular cartilage stability.

Nuclear factor kappa-B (NF-κB) activity within chondrocytes augments the expression of matrix degradation components, such as MMPs and ADAMTS-4, and fosters the generation of pro-inflammatory and destructive mediators, including COX-2 and iNOS [18,19]. The inhibitory impact of MT on the NF-κB signaling pathway has been consistently demonstrated [20,21]. Nuclear factor erythroid-derived 2-related factor 2 (Nrf2) is a transcription factor responsive to oxidative stress, overseeing the expression of diverse antioxidant enzymes, thereby contributing to cellular defense mechanisms [22,23]. Experimental findings have shown that mice with a knockout of the Nrf2 gene can suffer substantial cartilage damage [24]. The Nrf2/heme oxygenase-1 (HO-1) signaling pathway represents a central endogenous antioxidant system, serving as the principal defense mechanism against oxidative stress cytotoxicity [25,26]. The Nrf2/NADPH: Quinone Oxidoreductase 1 (NQO1) signaling pathway plays a key role in the development and progression of many diseases, including cancer, retinal aging, and others [27,28]. Additionally, activation of Nrf2 yields significant cartilage-protective effects, heightening the expression of downstream products such as HO-1 and NQO1, mitigating the secretion of IL-6, IL-10, and TNF-α, and retarding chondrocyte death or apoptosis [29]. Xinfeng Zhou et al. have reported the ability of MT to reduce reactive oxygen species and degradative enzymes by increasing the expression of Nrf2 and HO-1 [30]. 

Transforming growth factor-beta (TGF-β) holds considerable promise in fostering the formation, repair, and regeneration of cartilage cells [31]. Elevated levels of TGF-β 1/Smads serve to inhibit and ameliorate hypertrophy, apoptosis, necrosis, and autophagy in cartilage cells. Notably, bone morphogenetic proteins (BMPs) play pivotal roles as regulatory agents and effective inducers in cartilage formation, contributing to the proliferation and maturation of cartilage cells [32]. Previous studies have shown that inhibition of TGF-β signaling in subchondral bone mesenchymal stem cells can reduce OA [33]. Among them, BMP-2 is one of the main growth factors involved in cartilage regeneration. Research has found that BMP-2 induces cartilage formation in vitro through the expression of SOX9, Core binding factor-α (Runx 2), and downstream markers [34]. SOX9 assumes a critical regulatory role in cartilage formation, enhancing the synthesis of COL2A1 and proteoglycans in the ECM and thereby facilitating chondrocyte injury repair. Furthermore, SOX9 plays a key role in chondrocyte differentiation, development, and maturation [35]. Runx2 is indispensable for early cartilage formation, encompassing mesenchymal cell condensation, proliferation, and differentiation during early embryonic development [36]. BMP-7, in turn, promotes the synthesis of COL2A1 and proteoglycans in chondrocytes, augmenting the repair of damaged cartilage and safeguarding joint cartilage integrity [37,38]. Ming Pei et al. suggested that MT could stimulate chondrocyte synthesis by promoting the production of cartilage matrix components through the TGF-β signaling pathway [39]. Moreover, documented evidence supports the upregulation of SOX9, Runx2, and BMP2 by MT [40]. 

This research seeks to use SIRT1 as a focal point, delving into the inflammatory-linked NF-κB pathway, the oxidative Nrf2/HO-1 pathway, and the reparative TGF-β1/Smad2/BMPs pathway in comprehensive detail. This investigation strives to explore the ramifications of MT on pathological alterations, the expression of metabolic enzymes (MMPs and ADAMTs), and biomarker concentrations in both normal and OA rat knee joints. 

## 2. Results

### 2.1. In Vivo Experimental Results

#### 2.1.1. MT Alleviated Articular Cartilage Pathology

In the control group (Figure 1A), the cartilage structure remains relatively intact, exhibiting a smooth and regular surface with uniform chondrocyte distribution, and the cartilage matrix is appropriately stained in red. The model group (Figure 1B) displays irregular cartilage surfaces, chondrocyte loss, reduced cell numbers, and clustered distributions, with diminished red staining in the cartilage matrix. The MT group (Figure 1C) reveals a relatively intact cartilage structure with a smooth surface, minimal chondrocyte loss, and even distributions. Evaluation using the Mankin score (Appendix E) indicates a significant difference in cartilage scores between the model group, the control group, and the MT group (Figure 1D). These results show that the ACLT method successfully induces cartilage degeneration and that MT is effective in mitigating the degenerative process. 

#### 2.1.2. MT Reduced Synovial Tissue Inflammation

In the control group (Figure 2A), synovial tissue consists of a layer of cells with no inflammatory cell infiltration. The sub-synovial loose reticular connective tissue structure remains intact. In the model group (Figure 2B), there is substantial proliferation and hyperplasia of synovial cells, resulting in the formation of 2–3 layers. Some areas of the sub-synovial membrane show evident fibrosis and connective tissue proliferation, along with infiltration of inflammatory cells. In the MT group (Figure 2C), synovial cell proliferation is reduced, comprising only 1–2 layers. Some regions exhibit red blood cell extravasation, but there is no presence of inflammatory cells. This shows that MT can reduce the inflammatory reaction of synovial tissue and reduce tissue proliferation.

#### 2.1.3. MT Improves Subchondral Bone Lesions

In the control group (Figure 3A), the SBP maintains a relatively smooth interface with the adjacent cartilage. The STB exhibits a sponge-like structure with no evidence of fractures or defects, and mineralization appears uniformly distributed. Spindle-shaped osteocytes are evenly dispersed within the STB matrix. The bone marrow cavity contains a uniform density of bone marrow stromal cells, along with some adipose tissue, and a few osteoclasts are observable under high magnification. In the model group (Figure 3B), the SBP experiences a significant thinning, leading to a roughened interface with the cartilage. The STB density markedly decreases, resulting in a relative reduction in thickness. The number of bone cells significantly decreases, and they are sparsely distributed within the bone matrix. A substantial population of osteoclasts is visible under high magnification. In the MT group (Figure 3C), the SBP maintains a relatively smooth interface with the cartilage, and there is a significant increase in SBP thickness. The STB remains structurally intact with no signs of fractures or damage, while mineralization remains uniform, displaying a normal staining pattern for mineralized tissue. Osteocytes are uniformly distributed within the STB matrix, and the number of osteoclasts is significantly reduced under high magnification. 

#### 2.1.4. The Effects of MT on the Expression of Cartilage Metabolism-Related Proteins

To assess the impact of MT on the rat articular cartilage matrix and cell metabolism, this study conducted immunological analyses targeting MMP-3, MMP-13, ADAMTS-4, Aggrecan, receptor activator of nuclear factor-κB ligand (RANKL), and COL2A1 within the cartilage. ICH (Figure 4) revealed that the presence of positive cells related to cartilage matrix degradation, specifically RANKL, MMP3, MMP13, and ADAMTS-4, exhibited a significant increase in the model group (** *p* < 0.01). Conversely, the brown staining associated with the protective matrix proteins Aggrecan and COL2A1 saw a significant reduction in this model group (** *p* < 0.01). The MT group demonstrated a significant reduction in the number of positive cells for MMP3, MMP13, and ADAMTS-4 (** *p* < 0.01), along with a significant increase in the staining of Aggrecan and COL2A1 (** *p* < 0.01). 

#### 2.1.5. MT Reduces the Levels of TNF-α, COX-2, PGE2, and iNOS in Rat Serum

The ELISA results revealed that, in comparison to the model group, the concentration of MT in the serum of the MT group exhibited a significant increase (Figure 5A) (** *p* < 0.01). Furthermore, when compared to the control group, the model group displayed a significant increase in the levels of iNOS, COX-2, PGE2, and TNF-α within the serum (** *p* < 0.01). Conversely, the MT group exhibited a significant reduction in the levels of iNOS, COX-2, PGE2, and TNF-α in the serum when compared to the model group (Figure 5B–E) (** *p* < 0.01). 

### 2.2. In Vitro Experimental Results

#### 2.2.1. MT Activates SIRT1 and Inhibits the NF-κB Pathway in IL-1β-Induced Chondrocytes

WB and q-PCR techniques were employed to investigate the impact of MT on SIRT1 expression within chondrocytes. The results indicated that, in comparison to the model group, the MT group exhibited a substantial increase in SIRT1 expression in chondrocytes induced by IL-1β (Figure 6B,F) (** *p* < 0.01). Meanwhile, an in-depth exploration of the effects of MT on the NF-κB signaling pathway revealed that, relative to the model group, MT at concentrations of 100, 200, and 400 ng/mL significantly reduced the expression of phosphorylated NF-κB p65(p-p65) (*p* < 0.05), whereas MT at a concentration of 800 ng/mL significantly reduced its expression (*p* < 0.01) (Figure 6C). As per the WB results, all MT concentrations significantly lowered the expression of phosphorylated inhibitor of nuclear factor kappa-B alpha (p-IκBα) in chondrocytes (Figure 6E) (*p* < 0.01). Consequently, MT exhibits the potential to notably ameliorate the activation of the NF-κB pathway in IL-1β-induced chondrocytes. Further experiments are warranted to delve into the relationship between SIRT1 and the NF-κB pathway. 

#### 2.2.2. The Effects of MT on the Expression of Nrf1, HO-1, iNOS, and COX-2 in Chondrocytes

In our exploration of the role of MT as a comprehensive antioxidant in IL-1β-induced chondrocytes, the WB results indicated that, in comparison to the control group, both the model group and the MT group increased the protein expression level of HO-1 at concentrations of 100, 200, and 400 ng/mL (# *p* < 0.05) (Figure 7B), with 800 ng/mL of MT significantly elevating its expression level (## *p* < 0.01). Moreover, in contrast to the model group, the MT group exhibited a significant decrease in the expression levels of iNOS and COX-2 (Figure 7C–E) (** *p* < 0.01), as confirmed by q-PCR, which demonstrated consistent transcriptional changes for the COX-2 gene. Compared with the control group, the transcript levels of the Nrf2 gene were slightly elevated in the model group, and there were different degrees of elevation in the MT group, all of which were higher than the model group (Figure 7F) (## *p* < 0.01). 

#### 2.2.3. MT Promotes the Expression of TGF-β1/p-Smad2/BMPs in Chondrocytes

To delve into the mechanism behind the protective effect of MT on chondrocytes, we delved into the TGF-β1/Smad2/BMPs pathway. WB results (Figure 8B,C) clearly demonstrated that, when compared to the model group, MT led to a significant increase in the expression of TGF-β1 and p-Smad2 (** *p* < 0.01). The results of q-PCR (Figure 8D–G) further strengthened these findings by indicating that MT notably enhanced the gene transcription of BMP2, BMP7, SOX9, and Runx2. Consequently, we can reasonably infer that MT’s protective effect on chondrocytes is realized through modulation of the TGF-β1/p-Smad2/BMPs pathway. 

#### 2.2.4. In IL-1β-Induced Chondrocytes, MT Decreased MMP3, MMP13, and ADAMTS-4 and Increased COL2A1

To further substantiate the protective impact of MT on IL-1β-induced chondrocytes, we conducted an assessment of the expression levels of MMP3, MMP13, ADAMTS-4, and COL2A1. Western blot results (Figure 9A–E) demonstrated a significant reduction in the expression of MMP3, MMP13, and ADAMTS-4 in the MT group in comparison to the model group, while the expression of COL2A1 showed a substantial increase (** *p* < 0.01). Additionally, the q-PCR results (Figure 9F–I) reinforced these observations, showing that varying concentrations of MT led to a notable decrease in the transcriptional levels of MMP3, MMP13, and ADAMTS-4 and a concurrent increase in the transcriptional levels of COL2A1 to varying degrees.

#### 2.2.5. The Effect of the SIRT1 Inhibitor EX537 on Melatonin in Chondrocytes

##### EX527 Reverses the Effect of MT on the SIRT1 and NF-κB Pathways in IL-1β-Treated Chondrocytes

To delve deeper into the role of MT in regulating SIRT1 and associated signaling pathways, this experiment introduced the well-established SIRT1 inhibitor, EX527. The results of both WB and q-PCR (Figure 10B,F) clearly indicated that there was no significant change in the EX527 group compared with the model group and the EX527+MT group, which shows that the addition of EX527 significantly inhibited the expression of SIRT1. This pattern was consistently observed in the IF (Figure 10G) results as well. Notably, the WB findings revealed that, in contrast to the model group, there was no discernible difference in the ratios of p-p65/p65 and p-IκBα/IκBα between the EX527 group and the X527+MT group. 

##### The Effects of EX527 on HO-1, iNOS, COX-2, and Nrf2 in MT-Induced Chondrocytes

We investigated the functional interplay between SIRT1 and oxidative-related markers through WB and q-PCR. The experimental results indicated that, in comparison to the model group, the SIRT1 inhibitor EX527 significantly attenuated the expression of HO-1 (* *p* < 0.05) and Nrf2 (* *p* < 0.05) (Figure 11B,F) while exhibiting no significant influence on the expression levels of iNOS and COX-2 (Figure 11C–E). 

##### The Effects of EX527 on the TGF-β1/Smad2/BMPs Pathway in MT-Treated Chondrocytes

Our experiment has revealed that MT may protect cartilage via the TGF-β/p-Smad2 pathway, and SIRT1 appears to play a crucial role in mediating the effects of MT. To further investigate this, we introduced the SIRT1 inhibitor EX527. The results from WB analysis demonstrated that MT significantly increased the protein expression of TGF-β1 and p-Smad2 induced by IL-1β in chondrocytes (* *p* < 0.05) (Figure 12B,C,H). Interestingly, the inhibition of SIRT1 did not reverse the impact of MT on these proteins (* *p* < 0.05). Furthermore, q-PCR results indicated that there were no significant differences in the transcript levels of BMP2, BMP7, Runx2, and SOX9 between the EX527 group and the EX527+MT group compared to the model group (Figure 12D–G). 

##### The Effects of EX527 on the Expression of MMP3, MMP13, ADAMTS-4, and COL2A1 in MT-Treated Chondrocytes

We also conducted an investigation into the impact of inhibiting SIRT1 on the IL-1β-induced expression of MMP3, MMP13, ADAMTS-4, and COL2A1 in chondrocytes treated with MT (Figure 13). Both WB (Figure 13B–D) and q-PCR(Figure 13F–H) results showed that there was no difference between the EX527 group and the EX527+MT group compared to the model group. This shows that the addition of EX527 reversed the effects of MT on MMP3, MMP13, and ADAMTS-4. Our findings also extended to the study of COL2A1, showing that MT increased the expression of COL2A1 (Figure 13E,I,J). Nevertheless, the addition of EX527 continued to counteract this effect, as demonstrated by the IF results. 

## 3. Discussion

In this experiment, MT can mitigate chondrocyte degradation, reduce synovial tissue inflammation, restrain excessive osteoclast activation, protect the SBP and trabecular bone structure, and reduce the concentrations of inflammatory and oxidative mediators. In in vitro experiments, MT could decrease RANKL, MMP3, MMP13, and ADAMTS-4 in IL-1β-induced chondrocytes while enhancing the expression of aggrecan and COL2A1. RANKL is implicated in the regulation of chondrocyte dedifferentiation and apoptosis, with RANKL induced by hypertrophic chondrocytes contributing to osteoclast generation [41]. These findings indicate that MT can mitigate chondrocyte degradation, reduce synovial tissue inflammation, restrain excessive osteoclast activation, and protect the SBP and trabecular bone structure. Ming et al.’s experiments revealed that MT promotes ECM synthesis in chondrocytes, leading to increased production of COL2A1 and aggrecan [39]. ELISA analyses of serum further demonstrated that intraperitoneal injection of 30 mg/kg MT significantly elevated MT levels in the serum, while concurrently reducing the concentrations of inflammatory and oxidative mediators such as TNF-α, iNOS, COX-2, and PGE2. In vitro experiments corroborated that MT could decrease MMP3, MMP13, ADAMTS-4, iNOS, and COX-2 levels in IL-1β-induced chondrocytes while enhancing the expression of COL2A1. Earlier studies have demonstrated that MT can reduce TNF-α, IL-1β, and NO in the blood, consequently mitigating cartilage degradation [42,43]. Therefore, in vivo, MT effectively mitigates articular cartilage degeneration induced by ACLT surgery, simultaneously reducing the synovial inflammatory response and ameliorating subchondral bone lesions.

This study employed IL-1β in vitro to mimic an inflammatory and oxidative environment within chondrocytes [44]. It is well-established that SIRT1 serves as a pivotal mediator for several functions of MT [45,46,47]. In order to elucidate the mechanism of action of MT on chondrocytes, we sought to investigate the effect of MT on SIRT1. Experiments demonstrated that MT significantly increased the reduced SIRT1 levels induced by IL-1β in primary chondrocytes. Lim et al. argued that MT confers protective and anti-inflammatory effects through the SIRT1 pathway [11]. To further verify the role of MT in SIRT1, we introduced the SIRT1 inhibitor EX527. Both EX527 and IL-1β can reduce the levels of SIRT1, while MT can elevate the IL-1β-suppressed levels of SIRT1, with no impact on the inhibitory effect of EX527. This suggested that the protective effects of MT on chondrocytes may be mediated through SIRT1. 

To further elucidate the protective mechanism of MT, this experiment delved into the NF-κB and Nrf2/HO-1 pathways. Results demonstrated that MT significantly suppressed the expression of p-p65 and p-IκBα, while enhancing the expression of antioxidant markers Nrf2 and HO-1. Su Peiqiang et al. also noted that MT was resistant to IL-1β-mediated activation of the NF-κB pathway and inflammation in chondrocytes [48]. MT inhibited the nuclear translocation of NF-κB p65, as well as the production of reactive oxygen species and MMPs [2]. Earlier research has proposed that MT can elevate Nrf2 protein content by preventing its degradation and facilitating its translocation into the cell nucleus, thus reducing ECM synthesis in OA chondrocytes through the augmentation of Nrf2 and HO-1 protein levels [49]. It is evident that MT exerts anti-inflammatory and antioxidant effects in IL-1β-induced chondrocytes, possibly through the NF-κB and Nrf2/HO-1 pathways.

Is it possible for MT to modulate the NF-κB and Nrf2/HO-1 pathways through SIRT1? To test this hypothesis, we introduced the SIRT1 inhibitor EX527. The results of the experiment revealed that MT could counteract the impact of IL-1β on the NF-κB pathway and Nrf2/HO-1, but it could not alter the effects of EX527 on these pathways. This observation suggests that inhibition of SIRT1 reverses the effects of MT on these pathways, indicating that the effects of MT on the NF-κB pathway and Nrf2/HO-1 may be influenced by SIRT1. Previous research in the field of the nervous system has also shown that MT can restore SIRT1, NRF2, and HO-1 levels to normal when they have been reduced by LPS [50]. Additionally, our experiment uncovered that in the presence of EX527, MT was unable to reduce the expression of MMP3, MMP13, and ADAMTS-4, which are associated with cartilage ECM degradation induced by IL-1β. It also could not restore the expression of COL2A1. Therefore, it can be concluded that the protective effect of MT on chondrocytes is indeed mediated through SIRT1.

MT was found to increase the expression of COL2A1 in chondrocytes, as well as the thickness of the SBP, and reduce trabecular bone damage. In vitro experiments also demonstrated that MT augmented the expression of both COL2A1 protein and mRNA. Previous reports in the literature have shown that the combination of TGF-β1 and BMP2 can induce synovial ectopic tissue to differentiate into cartilage and positively influence BMP2-induced cartilage formation [51]. This study, therefore, explored the TGF-β1/Smad2/BMPs pathway through in vitro experiments. The results of WB and IF revealed that MT significantly increased the expression of TGF-β1 and p-Smad2, facilitating the translocation of p-Smad2 into the nucleus. The results also indicated that MT heightened the transcription of BMP2 and BMP7 mRNA. Research has indicated that MT can aid in the regeneration of degenerative intervertebral disc tissue in rats by activating the TGF-β1 pathway and the BMP/Smads pathway [52,53]. The importance of SOX9 and Runx2 in chondrocyte terminal differentiation and bone formation during late embryonic development is well-established [54]. This experiment, likewise, demonstrated that MT increased the transcription of SOX9 and Runx2 mRNA. In IL-1β-treated chondrocytes, the internal activation of the TGF-β pathway led to the activation of cartilage formation marker genes, including COL2a1, Acan, and SOX9 [55,56]. Furthermore, Kulkarni P’s research indicated that the NF-κB pathway could indirectly accelerate the degradation of chondrocyte ECM and lead to OA cartilage degradation by inhibiting SOX9 expression [57]. However, this experiment did not further explore the relationship between the NF-κB pathway and the TGF-β pathway, which is a deficiency of this experiment.

Moreover, our research uncovered that MT can activate the TGF-β1 and p-Smad2 pathways, and importantly, EX527 does not alter the impact of MT. This suggests that the influence of MT on the TGF-β1/p-Smad2 pathway is not mediated through SIRT1. However, inhibiting SIRT1 did alter the activating effects of MT on BMP2, BMP7, SOX9, and Runx2, suggesting a potential role for SIRT1 in the BMP pathways. The existing literature has indeed demonstrated that the activation of the SIRT1 pathway can enhance the gene expression of BMP7 in chondrocytes [58]. In addition, the increase in COL2A1 in chondrocytes may be related to MT activation of the TGF-β1/Smad2/BMPs pathway. Nevertheless, the addition of EX527 decreased COL2A1 levels, suggesting that the damage caused by the inhibition of SIRT1 outweighs the reparative effect of activating the TGF-β1/p-Smad2/BMPs pathway.

In our experiments, we revealed the ability of MT to protect subchondral bone through activation of the TGF-β1/Smad2/BMP2/BMP7 pathway. Both in vitro and in vivo studies emphasize the potential of MT to attenuate inflammatory and oxidative responses within chondrocytes, promoting chondrocyte and subchondral bone repair while attenuating matrix degradation. This provides essential theoretical support for its chondroprotective potential. 

## 4. Materials and Methods

### 4.1. Reagents and Antibodies

GAPDH, SIRT1, p65, p-p65, IκBα, p-IκBα, HO-1, iNOS, COX-2, TGF-β1, Smad2, p-Smad2, ADAMTS-4, COL2A1, and Aggrecan were procured from ABconal, Woburn, MA, USA. MMP3 and MMP13 were sourced from Affinity, Milwaukee, WI, USA, while RANKL was obtained from Absin, Keystone, CO, USA. MT (98% (TLC), M5250) was procured from Sigma, Marlborough, MA, USA. Antibodies, brands and product codes are shown in the Table 1.

### 4.2. Animal Ethics

A total of 30 healthy male Sprague Dawley (SD) rats (250–300 g) were used for in vivo experiments and were procured from the Liaoning Changsheng Experimental Animal Center. In the in vitro experiment, one male and two females were placed in each cage. After pregnancy, females were housed individually. In our study, 14- to 21-day-old rats were used as a source of chondrocytes. All rats are housed as required. All animal handling procedures adhered to the guidelines and regulations stipulated by the Animal Ethics Committee of Northeast Agricultural University. 

### 4.3. Experimental Animal Grouping and Surgery

A total of 30 male SD rats were subjected to random allocation into three groups: the Sham group, the Model group, and the MT group. In the Sham group, only the skin tissue was incised. The Model group utilized the anterior cruciate ligament transection (ACLT) method. The MT group received intraperitoneal injections of 30 mg/kg/2 days MT [59]. After a duration of 12 weeks, serum samples and knee joint specimens were collected.

### 4.4. Safranin O-Fast Green Staining and Immunohistochemistry (IHC)

IHC was used to characterize the expression of MMP-3, RANKL, MMP-13, ADAMTS-4, Aggrecan, and COL2A1 in chondrocyte ECM. Subsequently, data were analyzed using Image-Pro Plus version 6.0 software (Media Cybernetics, Rockville, MD, USA).

### 4.5. Enzyme-Linked Immunosorbent Assay (ELISA)

According to the manufacturer’s instructions, the levels of MT (H256-1-2, NJJCBIO, Nanjing, China), iNOS (A014-1-2, NJJCBIO, Nanjing, China), COX-2 (H200, NJJCBIO, Nanjing, China), PGE2 (MM-0068R1, MEIMIAN, Wuhan, China), and TNF-α (MM-0180R1, MEIMIAN, Wuhan, China) in rat serum were detected using an ELISA kit.

### 4.6. Isolation and Culture of Primary Chondrocytes

The isolation and culture methods of primary chondrocytes are shown in Appendix A. The Control group (treated with DMEM/F12 medium), the Model group (10 ng/mL IL-1β), the MT group (100, 200, 400, and 800 ng/mL MT, +10 ng/mL IL-1β), the Negative Control group (10 ng/mL IL-1β + 20 μM EX527) [60], and the Inhibitor group (10 ng/mL IL-1β + 800 ng/mL MT + 20 μM EX527) were isolated for a uniform duration of 24 h before subsequent experimental procedures.

### 4.7. Protein Extraction and Western Blot (WB)

Cellular proteins were obtained and preserved in the same way as before. Detailed steps are shown in Appendix B. The respective antibody dilutions used were as follows: SIRT1, p65, IκBα, HO-1, iNOS, COX-2, TGF-β1, p-Smad2, MMP3, MMP13, ADAMTS-4, GAPDH, 1:3000; p-p65, p-IκBα, p-Smad2, COL2A1, 1:2000.

### 4.8. RNA Extraction and Quantitative RT-PCR (q-PCR)

RNA extraction and q-PCR procedures are described in Appendix C. The relative expression levels of the cycle threshold (CT) were determined employing the 2^−ΔΔCT^ method. The primer sequences are shown in Table 2.

### 4.9. Immunofluorescence (IF)

The detailed immunofluorescence procedure is shown in Appendix D.

### 4.10. Data Analysis

These experiments were repeated a minimum of three times, and our findings are presented as the mean ± standard deviation (SD). Multiple comparisons were conducted using Dunnett’s test within the framework of one-way ANOVA. Statistical significance was established when the *p*-value was less than 0.05.

## 5. Conclusions

The results suggest that MT protects articular cartilage through anti-inflammation, inhibition of matrix-degrading enzyme secretion, and promotion of subchondral bone remodeling. Furthermore, in in vitro experiments, MT effectively mitigates IL-1β-induced matrix degradation in rat chondrocytes. It accomplishes this by activating the SIRT1/NF-κB/Nrf2/TGF-β1/p-Smad2/BMPs pathway (Figure 14). However, this study lacks in-depth exploration of the subchondral bone and bone remodeling. Further observation of subchondral bone changes through micro-CT is needed, and there is also a lack of inhibitors to validate the relevant signaling pathways in the animal model. In the subsequent animal experiments, it is essential to further investigate the effects of MT on subchondral bone and the impact of inhibiting SIRT1 on signaling pathways. Currently, there are increasingly more studies on MT, most of which highlight its powerful functions. Therefore, future studies should pay more attention to the effects of MT intake on endocrine functions, osteoporosis, and overall body functions.

## Figures and Tables

**Figure 1 ijms-25-06202-f001:**
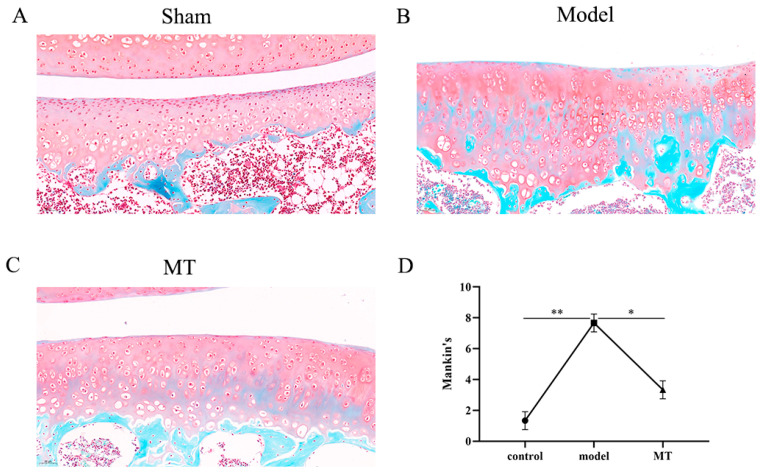
The effects of MT on pathological changes in articular cartilage (*n* = 3). Sham (only the skin tissue was incised), Model (using the anterior cruciate ligament transection (ACLT) method), and MT (30 mg/kg). (**A**–**C**) Safranin O-fast green staining was used to visualize pathological changes in articular cartilage in various groups (20×, 50 μm). (**D**) Mankin scoring. * *p* < 0.05, ** *p* < 0.01 (compared with the model group).

**Figure 2 ijms-25-06202-f002:**
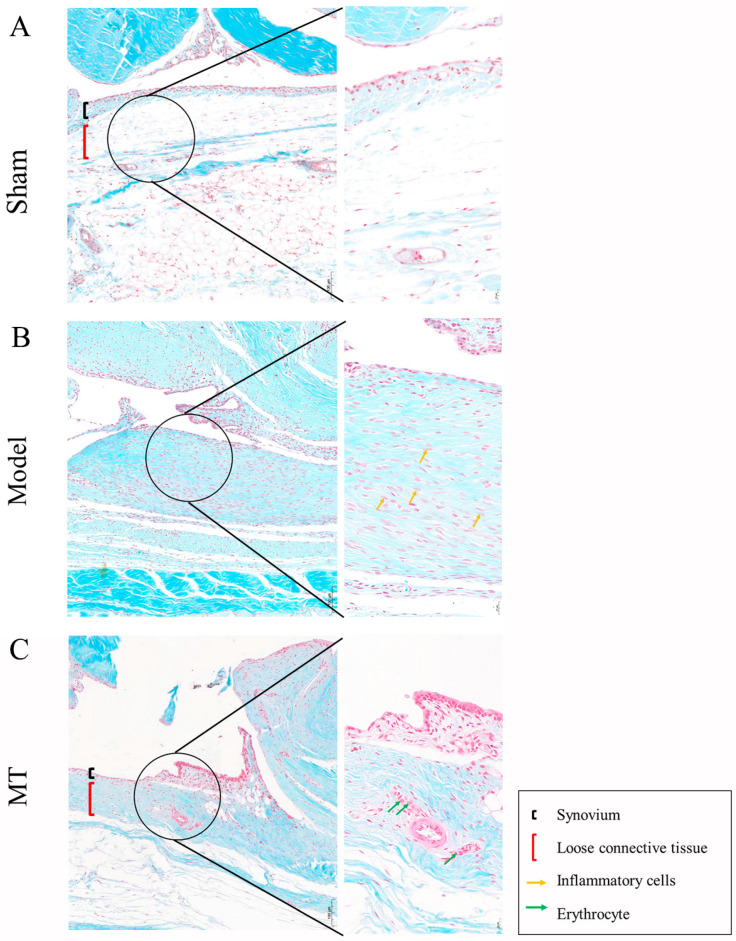
The effects of MT on synovial tissue (*n* = 3). Sham (only the skin tissue was incised), Model (using the anterior cruciate ligament transection (ACLT) method), and MT (30 mg/kg). (**A**–**C**) Safranin O-fast green staining was used to observe the status of synovial tissue (10×, 100 μm) (40×, 20 μm).

**Figure 3 ijms-25-06202-f003:**
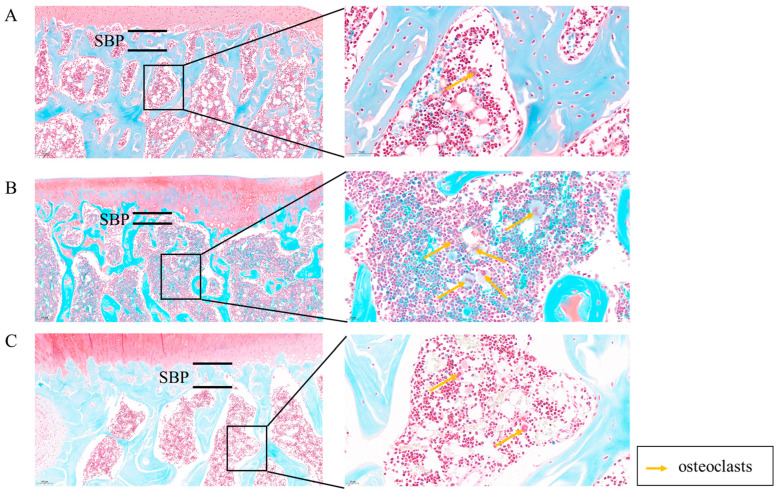
The effects of MT on subchondral bone (*n* = 3). Sham (only the skin tissue was incised), Model (using the anterior cruciate ligament transection (ACLT) method), and MT (30 mg/kg). (**A**–**C**) Safranin O-fast green staining was used to observe the subchondral bone status in each group (10×, 100 μm) (40×, 20 μm). (The subchondral bone plate, SBP).

**Figure 4 ijms-25-06202-f004:**
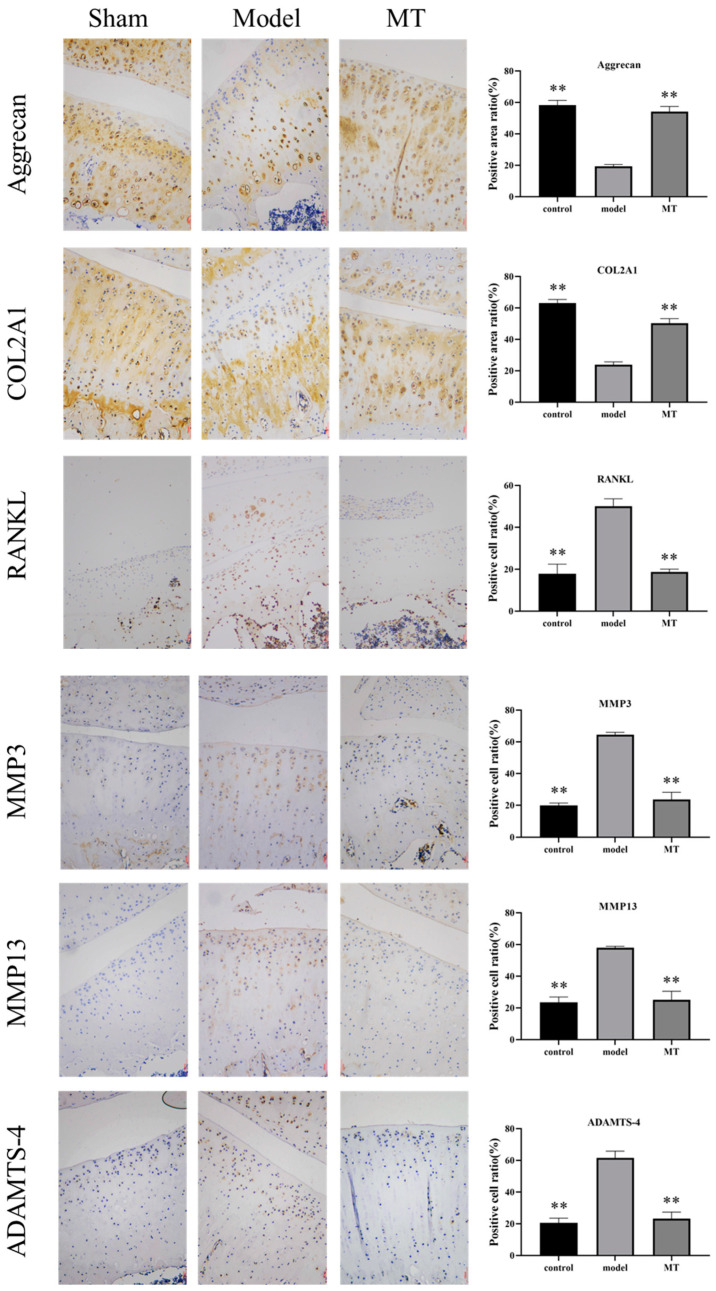
The effect of MT on the expression of cartilage metabolism-related proteins (*n* = 3). Sham (only the skin tissue was incised), Model (using the anterior cruciate ligament transection (ACLT) method), and MT (30 mg/kg). IHC staining was used to detect the content of Aggrecan, COL2A1, RANKL, MMP-3, MMP-13, and ADAMTS-4 in chondrocytes (100 μm). ** *p* < 0.01 (compared with the model group). (ICH: Immunohistochemistry; COL2A1: type II collagen; MMPs: matrix metalloproteinases; ADAMTS-4: A disintegrin and metalloprotease with thrombospondin motifs-4; RANKL: receptor activator of nuclear factor-κB ligand).

**Figure 5 ijms-25-06202-f005:**
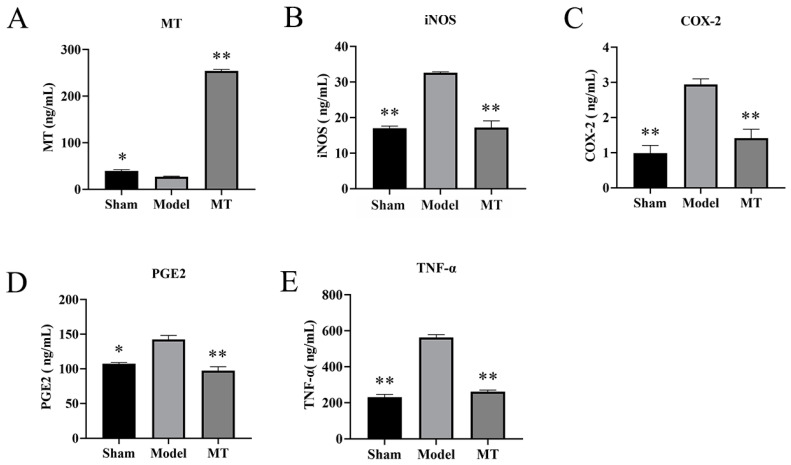
The effects of MT on the levels of MT, TNF-α, COX-2, PGE2, and iNOS in rat serum (*n* = 3). Sham (only the skin tissue was incised), Model (using the anterior cruciate ligament transection (ACLT) method), and MT (30 mg/kg). (**A**–**E**) The ELISA kits were used to detect the levels of MT, iNOS, COX-2, PGE2, and TNF-α in the serum of each group. All results are presented as mean ± standard deviation (SD) (*n* = 3), * *p* < 0.05, ** *p* < 0.01 (compared with the model group). (MT: Melatonin; iNOS: induced nitric oxide synthase; COX-2: Cyclooxygenase-2; PGE2: Prostaglandin E2; TNF-α: Tumor Necrosis Factor-α).

**Figure 6 ijms-25-06202-f006:**
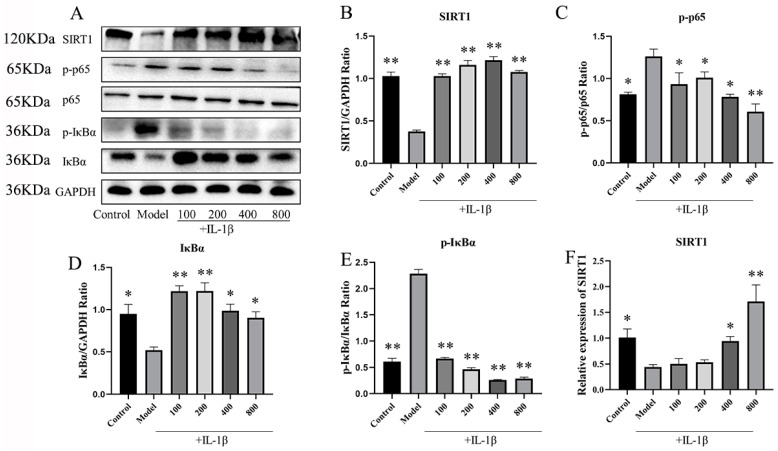
The effects of MT on SIRT1 and the NF-κB pathway (*n* = 3). (**A**–**E**) WB was used to detect the expression levels of SIRT1 and NF-κB pathway proteins. (**F**) q-PCR was used to detect the transcription levels of SIRT1 mRNA. All results are presented as mean ± standard deviation (SD) (*n* = 3), * *p* < 0.05, ** *p* < 0.01 (compared with the model group). (SIRT1: Silent information regulator transcript-1; p65: NF-κB p65; p-p65: phosphorylated NF-κB p65; IκBα: Inhibitor of nuclear factor kappa-B alpha; p-IκBα: phosphorylated inhibitor of nuclear factor kappa-B alpha; GAPDH: Glyceraldehyde-3-phosphate dehydrogenase).

**Figure 7 ijms-25-06202-f007:**
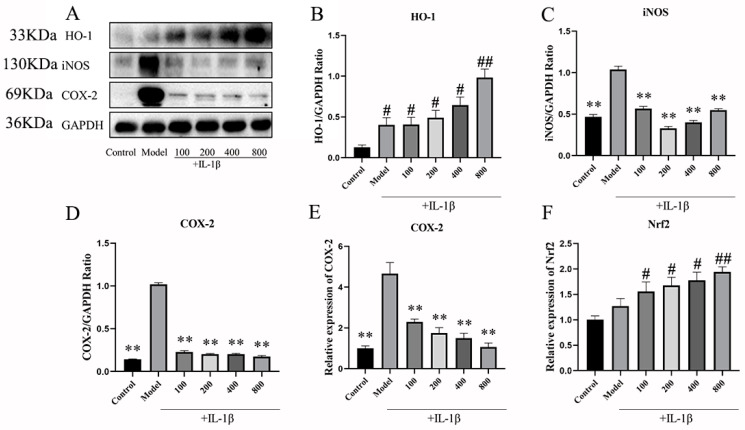
The effects of MT on HO-1, iNOS, and COX-2 (*n* = 3). (**A**–**D**) WB was used to detect the expression levels of HO-1, iNOS, and COX-2 pathway proteins. (**E**,**F**) q-PCR was used to detect the transcription levels of COX-2 and Nrf2 mRNA. All results are presented as mean ± standard deviation (SD) (*n* = 3), # *p* < 0.05, ## *p* < 0.01 (compared with the control group). ** *p* < 0.01 (compared with the model group). (HO-1: Heme Oxygenase 1; iNOS: induced nitric oxide synthase; COX-2: Cyclooxygenase-2; GAPDH: Glyceraldehyde-3-phosphate dehydrogenase).

**Figure 8 ijms-25-06202-f008:**
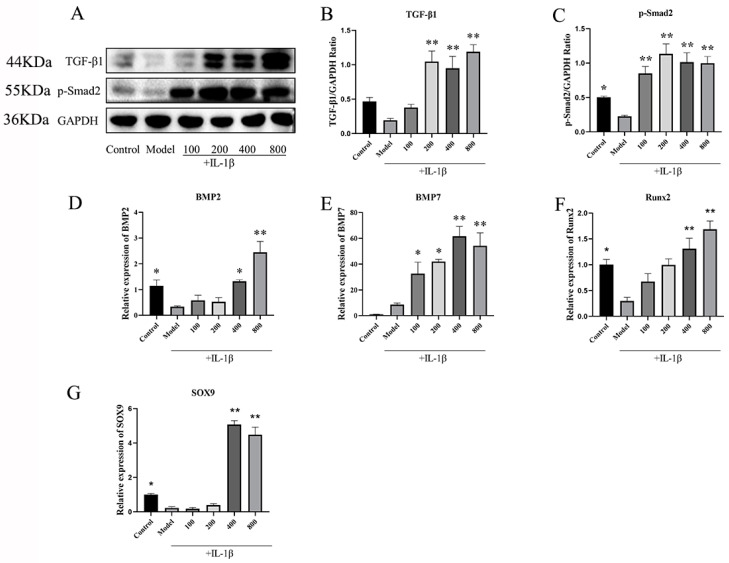
The effects of MT on the TGF-β1/p-Smad2/BMPs pathway (*n* = 3). (**A**–**C**) WB was used to detect the expression levels of TGF-β1 and p-Smad2 proteins. (**D**–**G**) q-PCR was used to detect the transcription levels of BMP2, BMP7, Runx2, and SOX9 mRNA. All results are presented as mean ± standard deviation (SD) (*n* = 3), * *p* < 0.05, ** *p* < 0.01 (compared with the model group). (TGF-β: Transforming growth factor-beta; p-Smad2: Phosphorylated mothers against decapentaplegic homolog 2; GAPDH: Glyceraldehyde-3-phosphate dehydrogenase).

**Figure 9 ijms-25-06202-f009:**
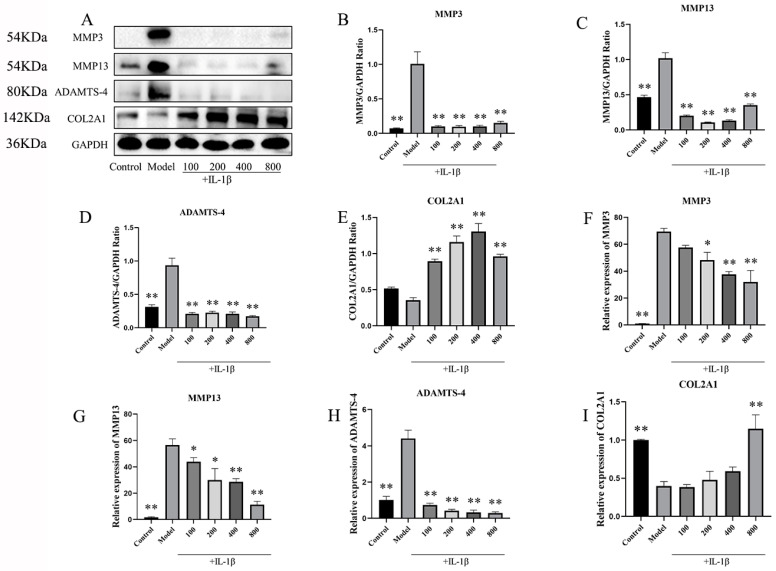
The effects of MT on the expression of MMP3, MMP13, ADAMTS-4, and COL2A1 (*n* = 3). WB (**A**–**E**) and q-PCR (**F**–**I**) were used to detect the levels of MMP3, MMP13, ADAMTS-4, and COL2A1 proteins and mRNA. All results are presented as mean ± standard deviation (SD), * *p* < 0.05, ** *p* < 0.01 (compared with the model group). (MMP3: matrix metalloproteinases-3; MMP13: matrix metalloproteinases-13; ADAMTS-4: A disintegrin and metalloprotease with thrombospondin motifs-4; COL2A1: type II collagen; GAPDH: Glyceraldehyde-3-phosphate dehydrogenase).

**Figure 10 ijms-25-06202-f010:**
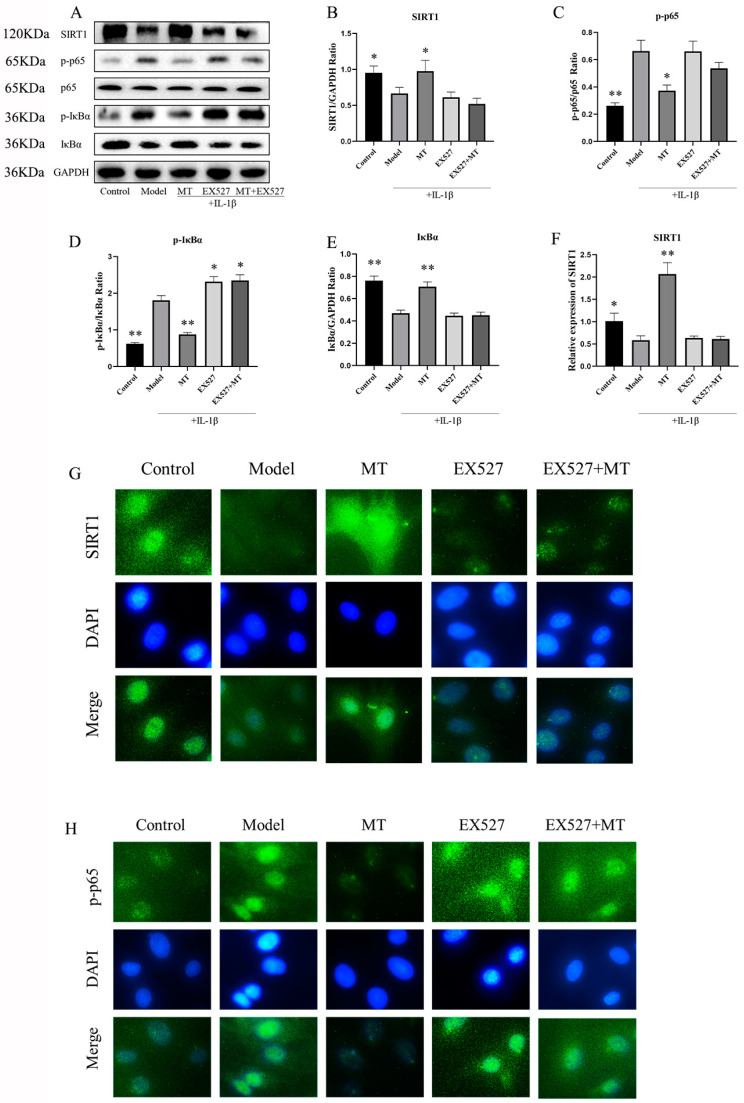
The effects of EX527 on the SIRT1 and NF-κB pathways in MT-treated chondrocytes (*n* = 3). (**A**–**E**) WB was used to detect the expression levels of SIRT1, p-p65, p-IκBα, and IκBα proteins. q-PCR (**F**) was used to detect the transcriptional levels of SIRT1 mRNA. IF (**G**,**H**) was used to detect the expression of SIRT1 and p-p65 in the nucleus (×400). All results are presented as mean ± standard deviation (SD) (*n* = 3), * *p* < 0.05, ** *p* < 0.01 (compared with the model group). (SIRT1: Silent information regulator transcript-1; p65: NF-κB p65; p-p65: phosphorylated NF-κB p65; IκBα: Inhibitor of nuclear factor kappa-B alpha; p-IκBα: phosphorylated inhibitor of nuclear factor kappa-B alpha; EX527: Selisistat; GAPDH: Glyceraldehyde-3-phosphate dehydrogenase).

**Figure 11 ijms-25-06202-f011:**
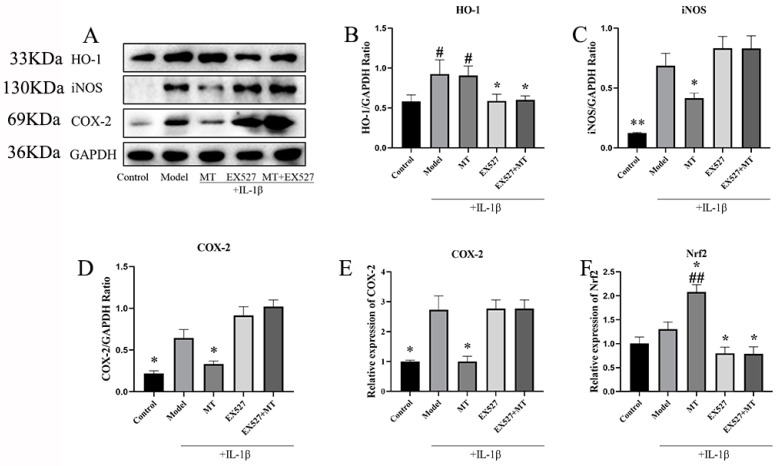
The effects of EX527 on HO-1, iNOS, COX-2, and Nrf2 in MT-induced chondrocytes (*n* = 3). (**A**–**D**) WB was used to detect the expression levels of HO-1, iNOS, and COX-2 proteins. q-PCR (**E**,**F**) was used to detect the transcript levels of COX-2 and Nrf2 mRNA. All results are presented as mean ± standard deviation (SD) (*n* = 3), # *p* < 0.05, ## *p* < 0.01 (compared to the control group). * *p* < 0.05, ** *p* < 0.01 (compared with the model group). (HO-1: Heme Oxygenase 1; iNOS: induced nitric oxide synthase; COX-2: Cyclooxygenase-2; EX527: Selisistat; GAPDH: Glyceraldehyde-3-phosphate dehydrogenase).

**Figure 12 ijms-25-06202-f012:**
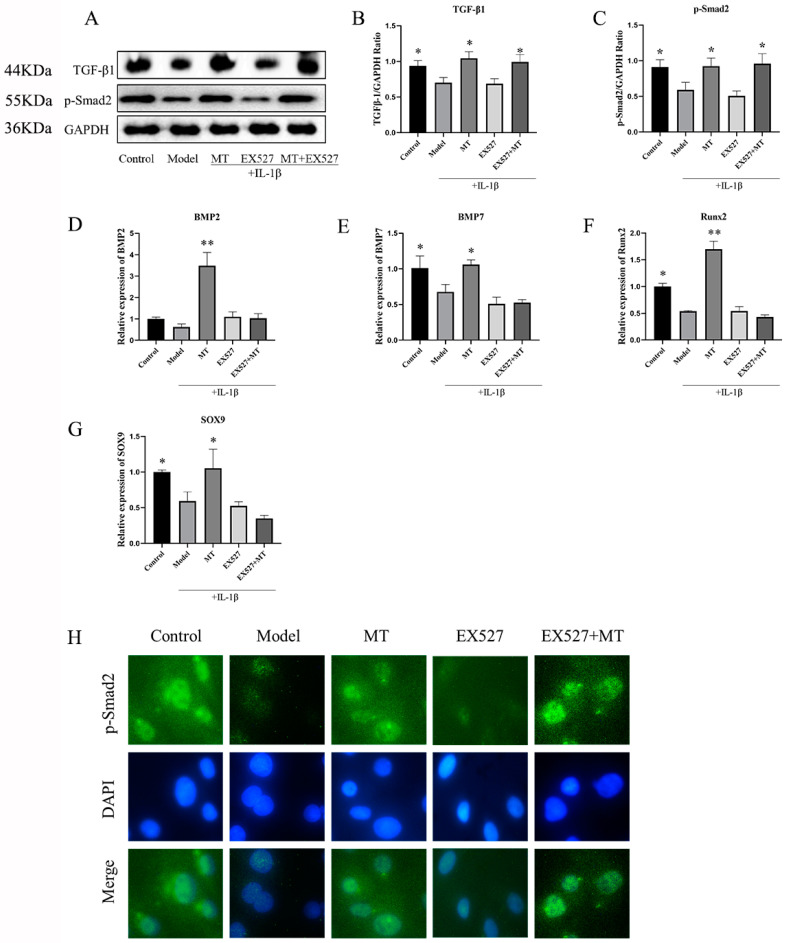
The effect of EX527 on the TGF-β1/Smad2/BMPs pathway in MT-treated chondrocytes (*n* = 3). (**A**–**C**) WB was used to detect the protein expression levels of TGF-β1 and p-Smad2. q-PCR (**D**–**G**) was used to detect the transcript levels of BMP2, BMP7, Runx2, and SOX9 mRNA. IF (**H**) was used to detect the expression of p-Smad2 in the nucleus (×400). All results are presented as mean ± standard deviation (SD) (*n* = 3), * *p* < 0.05, ** *p* < 0.01 (compared with the model group). (TGF-β: Transforming growth factor-beta; p-Smad2: Phosphorylated mothers against decapentaplegic homolog 2; EX527: Selisistat; GAPDH: Glyceraldehyde-3-phosphate dehydrogenase).

**Figure 13 ijms-25-06202-f013:**
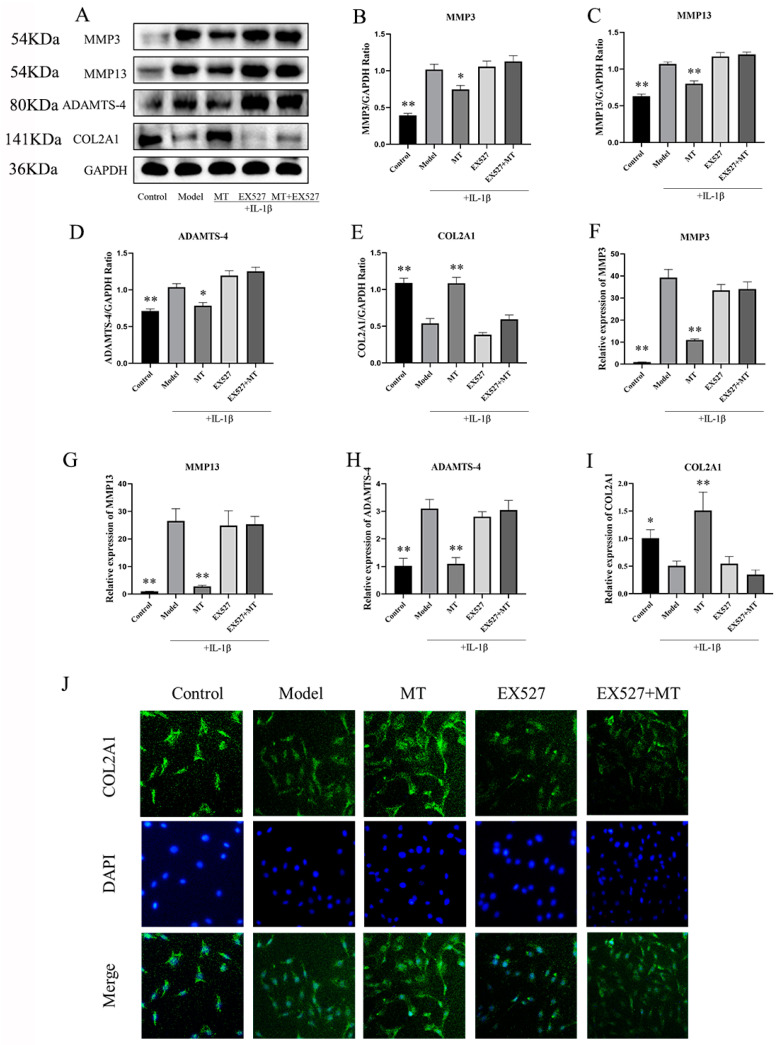
The Effects of EX527 on the expression of MMP3, MMP13, ADAMTS-4, and COL2A1 in MT-treated chondrocytes (*n* = 3). WB (**A**–**E**) and q-PCR (**F**–**I**) were used to detect the expression levels of MMP3, MMP13, ADAMTS-4, and COL2A1. IF (**J**) detects COL2A1 expression in chondrocytes (×200). All results are presented as mean ± standard deviation (SD), * *p* < 0.05, ** *p* < 0.01 (compared with the model group). (MMP3: matrix metalloproteinases-3; MMP13: matrix metalloproteinases-13; ADAMTS-4: A disintegrin and metalloprotease with thrombospondin motifs-4; COL2A1: type II collagen; EX527: Selisistat; GAPDH: Glyceraldehyde-3-phosphate dehydrogenase).

**Figure 14 ijms-25-06202-f014:**
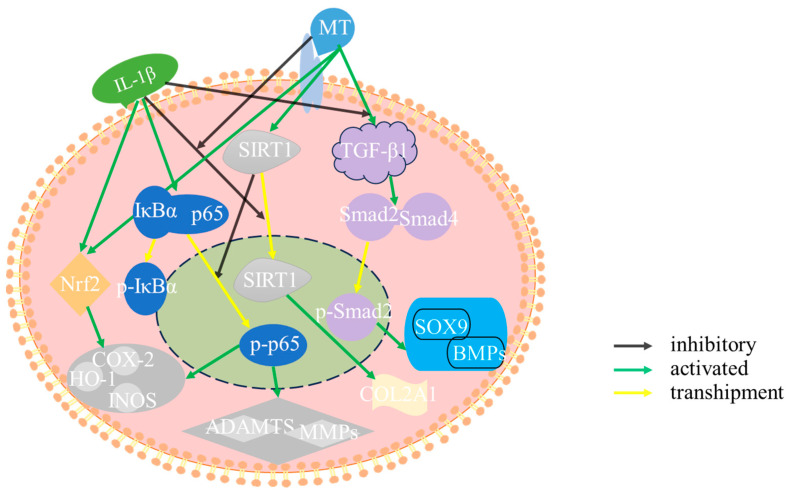
Mechanism diagram.

**Table 1 ijms-25-06202-t001:** Antibodies, brands, and product codes.

Antibodies	Brands	Product Codes
SIRT1	ABconal	A11267
MMP3	Affinity	AF0217
MMP13	Affinity	AF5355
ADAMTS-4	ABconal	A2525
COX-2	ABconal	A1253
Nrf2	ABconal	A0674
COL2A1	ABconal	A1560
HO-1	ABconal	A1346
iNOS	ABconal	A3774
TGF-β1	ABconal	A2124
p-Smad2	ABconal	AP0269
p65	ABconal	A2547
p-p65	ABconal	AP0123
IκBα	ABconal	A24742
p-IκBα	ABconal	AP0614
Aggrecan	ABconal	A8536
RANKL	Absin	abs120177
GAPDH	ABconal	AC001

**Table 2 ijms-25-06202-t002:** Primer sequence.

	Forward Primer	Reverse Primer
SIRT1	5′-TCAGTGTCATGGTTCCTTTGC-3′	5′-AATCTGCTCCTTTGCCACTCT-3′
MMP3	5′-TTTGGCCGTCTCTTCCATCC-3′	5′-GCATCGATCTTCTGGACGGT-3′
MMP13	5′-TTCTGGTCTTCTGGCACACG-3′	5′-TGGAGCTGCTTGTCCAGGT-3′
ADAMTS-4	5′-CACCGAACCGACCTCTTCAA-3′	5′-GAGTTCCATCTGCCACCCGT-3′
COX-2	5′-AGAAGCGAGGACCTGGGTTCAG-3′	5′-ACACCTCTCCACCGATGACCTG-3′
Nrf2	5′-CGAGATATACGCAGGAGAGGTAAGA-3′	5′-GCTCGACAATGTTCTCCAGCTT-3′
COL2A1	5′-GAACGGCGGCTTCCACTTCAG-3′	5′-CACACGCTAGACACTGGACT-3′
BMP2	5′-GAGGAGAAGCCAGGTGTCT-3′	5′-GTCCACATACAAAGGGTGC-3′
BMP7	5′-AAACAACGCAGCCAGAACCG-3′	5′-CCTCACAGTAGGCAGCATAGC-3′
SOX9	5′-CTCCCAAAACAGACGTGCAA-3′	5′-CGAAGGTCTCGATGTTGGAGAT-3′
Runx2	5′-TGCTGGAGTGATGTGGTTTTCT-3′	5′-CCCCTGTTGTGTTGTTTGGTAA-3′
GAPDH	5′-GATGCCCCCATGTTTGTGAT-3′	5′-GGCATGGACTGTGGTCATGAG-3′

## Data Availability

The original contributions presented in the study are included in the article, further inquiries can be directed to the corresponding author/s.

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
