# Peer review of "Melatonin Delays Arthritis Inflammation and Reduces Cartilage Matrix Degradation through the SIRT1-Mediated NF-κB/Nrf2/TGF-β/BMPs Pathway"

_ijms, 2024, doi:10.3390/ijms25116202_

Round 1

Reviewer 1 Report (Previous Reviewer 1)

Comments and Suggestions for Authors

I would like to thank the authors for addressing my initial comments. The authors have very effectively addressed all comments. Following the revision to the article, I feel that this manuscript is now acceptable for publication.

Author Response

Dear reviewer:

Thank you very much for identifying the weaknesses and omissions in this article, and for making it more rigorous and suitable for publication under your guidance. Thank you again for your guidance. I wish you all the best.

Reviewer 2 Report (Previous Reviewer 2)

Comments and Suggestions for Authors

The authors have revised their manuscript carefully and can be aceepted in the present form.

Author Response

Dear reviewer:

Thank you very much for identifying the weaknesses and omissions in this article, and for making it more rigorous and suitable for publication under your guidance. Thank you again for your guidance. I wish you all the best.

Reviewer 3 Report (New Reviewer)

Comments and Suggestions for Authors

Paper entitled: “Melatonin delays arthritis inflammation and reduces cartilage matrix degradation through the SIRT1-mediated NF-κB/Nrf2/TGF-β/BMPs pathway.” (ijms- 3039144). The subject of the study is very interesting and requires deepening the poorly understood impact of melatonin on arthritis inflammation reduction. The presented results are interesting, but some issues should be addressed before publication in IJMS:

L27: „Enzyme-linked immunosorbent assay (ELISA) results” - the explanation of the abbreviation should be higher on L 20.

There are other unexplained notations in the abstract such as Sham, Model

L44: Tumor Necrosis Factor - please write in lowercase letters

All descriptions of the figures lack an explanation of specific names.

L173: “These findings indicated that MT has a protective effect 173 on cartilage and mitigates the degradation of the cartilage matrix” - Such summaries should be included in the discussion or conclusion.

Figure 6, 7. the texts on the charts merge and therefore look bad graphically: CONTROL MODEL

Figure7 A) – HO-1 band for 800/IL-1β leaves much to be desired

It is not possible to view Fig.9 in its entirety

Fig. 10F) is different from the other charts

Fig. 9, 10, 11 - the description of EX527 shortcuts is missing

I think it is extremely important to point out the limitations that apply to work. This is missing from work, please fill it in.

Author Response

Dear reviewer:

Thank you very much for identifying the shortcomings and mistakes in this article, and for making it more rigorous under your guidance. I have finished revising all your suggestions, and the revised parts are marked with yellow color in the article. In addition, I have made a reply in this document, please check it. Thank you again for your guidance. I wish you all the best.

L27: “Enzyme-linked immunosorbent assay (ELISA) results” - the explanation of the abbreviation should be higher on L 20.

There are other unexplained notations in the abstract such as Sham, Model

I have adjusted the position of the ELISA as the reviewer's comments and explained Sham as well as Model in the abstract.

L44: Tumor Necrosis Factor - please write in lowercase letters

I've changed the Tumor Necrosis Factor to lowercase letters.

All descriptions of the figures lack an explanation of specific names.

I've added a description of all the specific names in the legend and provided a separate reconciliation of full names and abbreviations for specific names.

L173: “These findings indicated that MT has a protective effect 173 on cartilage and mitigates the degradation of the cartilage matrix” - Such summaries should be included in the discussion or conclusion.

Many thanks to the reviewers for their guidance on inappropriate expressions in the results, I have removed such expressions from the results as suggested

Figure 6, 7. the texts on the charts merge and therefore look bad graphically: CONTROL MODEL

Thanks to the reviewer's care, I have readjusted all the images to look clearer.

Figure7 A) – HO-1 band for 800/IL-1β leaves much to be desired

I replaced the current strip with a spare strip.

It is not possible to view Fig.9 in its entirety.

Thanks to the careful reviewers who caught this omission, I've adjusted the image to make sure it can be displayed in full!

Fig. 10F) is different from the other charts

I readjusted this image to make it consistent with the other images

Fig. 9, 10, 11 - the description of EX527 shortcuts is missing

I've added a description of EX527 to the legend.

I think it is extremely important to point out the limitations that apply to work. This is missing from work, please fill it in.

Many thanks to the reviewers for the shortcomings and I have added the limitations of this paper in the conclusion section.

Reviewer 4 Report (New Reviewer)

Comments and Suggestions for Authors

In this study authors evaluated the role of MT on chondrocytes and found that MT treatment reduced chondrocyte loss, synovial hyperplasia, and increased subchondral bone thickness. Moreover, MT down-regulated the expression of proteins involved in matrix degradation and reduced serum inflammatory cytokine levels  by upregulating the expression of SIRT1/Nrf2/TGF-β/BMPs while inhibiting the NF-κB pathway. 

the manuscript is interesting and generally well written but presents several flaws that must be resolved. In particular:

Line 10: Remove "Correspondence"

Lines 80-85: Since NRF2/KEAP1 signaling pathway plays a key role in this manuscript, its multifaceted role deserves to be highlighted. In fact, this signaling plays a key role in the onset and progression of several cancerous and non-cancerous diseases (see PMID: 37175546  and PMID: 37371607 ).

Figures legend: Please add the number of replicates (N) in the legend of each figure. 

Figure 1: Scale bars must be added

Figure 2: Images are blurry, please improve them 

Figure 4: Image quality is very low and tissue morphology is not appreciable

Figure 8A and 12A: Why authors did not evaluated total SMAD2 protein levels? alterations in pSMAD2 could be due to alterations in total SMAD2.  

Figure 9: Figure is cut on right side

Figure 13: the quality of IF images is very low since there is a lot of background in the green channel (COL2A1). In fact, you can see green outside of the cells

When western blot images are shown, molecular weights must be added

4.1. Reagents and Antibodies: a table reporting the antibodies used, product codes and dilutions must be added

4.5. Enzyme-Linked Immunosorbent Assay (ELISA): product codes are needed

4.9. Immunofluorescence (IF): Annex 4 is not present 

Abbreviations must be written in full length when mentioned for the first time

Author Response

Dear reviewer:

Thank you very much for identifying the shortcomings and mistakes in this article, and for making it more rigorous under your guidance. I have finished revising all your suggestions, and the revised parts are marked with yellow color in the article. In addition, I have made a reply in this document, please check it. Thank you again for your guidance. I wish you all the best.

Line 10: Remove "Correspondence"
I've removed "Correspondence".

Lines 80-85: Since NRF2/KEAP1 signaling pathway plays a key role in this manuscript, its multifaceted role deserves to be highlighted. In fact, this signaling plays a key role in the onset and progression of several cancerous and non-cancerous diseases (see PMID: 37175546IF: 5.6 Q1 and PMID: 37371607IF: 4.7 Q1 ).
Many thanks to the reviewers for their guidance in remedying the shortcomings of this paper, I have added " The Nrf2/NQO1 signaling pathway plays a key role in the development and progression of many diseases, including cancer, retinal aging and others [27,28]." added in the introduction and introduced references as required.

Figures legend: Please add the number of replicates (N) in the legend of each figure. 
I added the number of repetitions to the legend of each figure (n=3)

Figure 1: Scale bars must be added
Thank you very much to the reviewers for their carefulness in finding the errors in the details of the article. Scale bars are present in the lower left corner of the image, which may be unreadable due to the shrinking of the image. I have provided all the individual clear images in the attachment.

Figure 2: Images are blurry, please improve them 
Many thanks to the reviewer for pointing out the problem and I have readjusted the clarity of the image. Just in case the image was compressed and caused it to be unclear, I am submitting the image again separately via an attachment.

Figure 4: Image quality is very low and tissue morphology is not appreciable
I have readjusted the clarity of the image. Just in case the image was compressed and caused it to be unclear, I am submitting the image again separately via an attachment.

Figure 8A and 12A: Why authors did not evaluated total SMAD2 protein levels? alterations in pSMAD2 could be due to alterations in total SMAD2.  
Dear reviewer, at the initial stage of experimental design, we believed that pSMAD2 is a key regulator that enters the nucleus to initiate transcription and regulate gene expression. We planned to demonstrate that an increase in pSMAD2 levels and more p-SMAD entering the nucleus can support our hypothesis without specifically testing total SMAD2 levels. However, it appears that our initial assumption may not be mature enough. In future experimental designs, we will continuously refine our approach to make the experiments more rational. Thank you for your understanding and guidance.

Figure 9: Figure is cut on right side
Thanks to the careful reviewers who caught this omission, I've adjusted the image to make sure it can be displayed in full!

Figure 13: the quality of IF images is very low since there is a lot of background in the green channel (COL2A1). In fact, you can see green outside of the cells.
I have readjusted the background of COL2A1 as per the reviewer's comments. It is possible that the extracellular green spots are impurities left over from the experimental process and reagents added to prevent fluorescence quenching.

When western blot images are shown, molecular weights must be added
I have added molecular weights to each western blot image.

4.1. Reagents and Antibodies: a table reporting the antibodies used, product codes and dilutions must be added
Many thanks to the reviewers for their suggestions that remedied the shortcomings of this article, I have added a table about the antibodies and product codes as requested, but since the dilutions used in each experiment are different, I have added the dilutions of the required antibodies in the place corresponding to each experimental method.

4.5. Enzyme-Linked Immunosorbent Assay (ELISA): product codes are needed
I added product codes for each ELISA product

4.9. Immunofluorescence (IF): Annex 4 is not present 
I've added the full attachment to the conclusion.

Abbreviations must be written in full length when mentioned for the first time.
I went through the text and made changes to ensure that the first reference to a scientific term was written with the full name, except in the abstract. In addition, I have provided a separate reconciliation of the full names and abbreviations of specific names in the attached appendix.

Round 2

Reviewer 4 Report (New Reviewer)

Comments and Suggestions for Authors

the manuscript has been significantly improved and can be accepted in the present form 

This manuscript is a resubmission of an earlier submission. The following is a list of the peer review reports and author responses from that submission.

Round 1

Reviewer 1 Report

Comments and Suggestions for Authors

1- Several information and citations are missing in the introduction section. The introduction section should be updated by data more related and updated data from Scopus.

2. The clinical evidences should be provided for following sentence.

Melatonin possesses anti-inflammatory properties, regulates bone metabolism, inhibits osteoclasts, and exerts antioxidant effects

3. The emphasis should be given on the discussion of the results like why certain effects are coming in to existence and what could be the possible reason behind them?

4. The results lack novelty.

5. The discussion needs to be written in more detail.

6- The conclusion section is needed to be rewritten by including future perspectives.

7. This topic is very important. But some paper has been published with this context. I request to write a few sentences about the perspective.

8. Are there any recommendations provided around optimal patient selection, dosage regimen, for potential areas of further research needed?

Comments on the Quality of English Language

Grammatical errors should be rectified.

Reviewer 2 Report

Comments and Suggestions for Authors

The manuscript titled “Melatonin delays arthritis inflammation and reduces cartilage matrix degradation through the SIRT1-mediated NF-κB/Nrf2/TGF-β/BMPs pathway” investigated the effect of melatonin on delaying arthritis inflammation and reducing cartilage matrix degradation in vivo and in vitro experiments. The authors also investigate its regulator mechanism via the SIRT1-mediated NF-κB/Nrf2/TGF-β/BMPs pathway. However, some revisions should be modified as follows:

1.     In the title, there should be no full stop in the title.

2.     In the section of introduction, the introduction about SIRT1 (Line 49-51) needed to refer to PMID: 37308008; the introduction about TGF-β (Line 67-68) needed to refer to PMID: 23685840.

3.     Using the abbreviation for melatonin (MT) in Line 48, and use the MT for melatonin in the following paper.

4.     The TNF-α, IL-1β, and other abbreviation needed to be used the full name for the first time.

5.     In the experiment, why the authors just used a single dose of MT? How to choose this dosage, and why not establish a control + MT group to check the influence of MT on normal mice?

6.     In the experiment, why the authors not use the positive drugs?

7.     The authors needed to improve their language native, such as Line 223 (delve into), Line 231 (WB indicated that, both IL-1β……).

8.     The figures in this paper were not clear enough, the authors needed to improve the figures clearly (Figure 4-13).

9.     The defects of this paper or further studies needed to be reflected in conclusion.

10.   An associated mechanism diagram is missing, the authors can draw a full-text mechanic diagram and put it in the section of conclusion or abstract graphic.

Comments on the Quality of English Language

The authors needed to improve their language native, such as Line 223 (delve into), Line 231 (WB indicated that, both IL-1β……).